# Effect of flipped classroom approach in the teaching of a hematology course

Haitham Qutob [1,2] *

1 Faculty of Applied Medical Sciences, Department of Medical Laboratory Technology, King Abdulaziz University, Rabigh, Saudi Arabia, 2 Medical Laboratory Sciences Department, Fakeeh College for Medical Sciences, Jeddah, Saudi Arabia

* hqutob@kau.edu.sa

**Data Availability Statement:** All relevant data are within the manuscript, its Supporting Information files, and https://figshare.com/s/17fcd1aabcc8a2b35f8e.

**Funding:** The author(s) received no specific funding for this work.

## Abstract

The flipped classroom is a teaching strategy used to enhance the engagement of students in the learning process. Replacing didactic lectures with active learning strategies through flipped classroom sessions fosters independence and the opportunity for students to engage in different passive learning styles. Although many studies of the flipped classroom approach have been conducted with medical students, few have focused on the effect of flipped classroom sessions on students in other medical professional fields. The aim of this study was to assess the effect of the flipped classroom approach on the performance of medical laboratory science students in a hematology course and their perception of the flipped classroom as an active learning strategy. In addition, students' perceptions regarding the flipped classroom as an active learning strategy were assessed. Of two cohorts in hematology courses, cohort 2 attended flipped classroom sessions, whereas cohort 1 underwent traditional class sessions. Students' learning outcomes, achievements and performance on examinations in hematology courses were assessed. In addition, a satisfaction survey was distributed to the students to evaluate their perception of the flipped classroom as a student-centred learning strategy. Students in cohort 2 performed significantly better in the knowledge and cognitive domains than did those in cohort 1 ($p < 0.001$). Cohort 2 students completed the course successfully with an average grade of 81%, and few students received low grades; in comparison, cohort 1 students completed the course with an average grade of 73%, and 7 students received a grade of F. Of students in cohort 2, 83% believed that the flipped classroom provided a better understanding of the subject matter and appropriate knowledge and skills. The results indicate the effectiveness of flipped classrooms as an active learning style in enabling students to obtain desirable knowledge and improve their academic performance. Moreover, students zrecognized that the flipped classroom as an active learning style was more beneficial than the traditional teaching approach.

## Introduction

In recent decades, new teaching strategies in higher education have been implemented. Many medical colleges have switched from traditional didactic methods to student-centred learning

**Competing interests:** The authors have declared that no competing interests exist.

[1,2], in which students are encouraged to participate, engage and identify the appropriate resources and needs to fulfil the required objectives [2]. This kind of learning strategy motivates and enables students to be life-long learners, fosters self-growth and enables retention of up-to-date information about new advances and approaches in various medical fields [3]. Nowadays, national and international accreditation agencies must incorporate active learning strategies in medical programs to provide students the skills and characteristics of life-long learners [4].

An active learning style has shown a positive effect in comparison with traditional teaching [1]. The flipped classroom approach, case-based learning, problem-based learning and blended learning styles are new teaching strategies involving student-centred learning that enable students to cultivate interpersonal skills, obtain medical knowledge and improve cognition [5]. In the flipped classroom approach, students have the study materials and prepare for class before the class itself, and in the class, they share basic knowledge and build upon the concepts taught according to the set objectives [6]. Several studies have confirmed that student-centred learning as an active learning strategy is associated with improved student performance, reduced failure rate and better learning achievements [1,7–9]. A study conducted with students taking an anatomy course revealed that the achievement of course objectives was significantly improved when the course was delivered through an active teaching method than when delivered in the traditional method [10]. Moreover, the students in the study stated that that the active teaching method helped them gain knowledge, skills and confidence on examinations [10]. Another study with students in a microbiology course demonstrated that problem-based learning, a student-centred learning strategy, helped students retain information and improve cognitive skills [11].

Many other studies have shown that the flipped classroom has a positive effect on student achievement and performance [3,6]. In an obstetrics and gynaecology course, for example, students taught with the flipped classroom approach had better performance and test scores than did those who were taught in a traditional method [12]. Arya et al., in addition, demonstrated that students in the flipped classroom showed better academic performance in complete medical courses than did students in traditional teaching classrooms [13]

On the other hand, some studies have shown no significant difference in examination scores between students in the flipped classroom and those receiving didactic teaching, although the students' perception about the flipped classroom and blended learning was positive [14]. The reasons for the findings in those studies may include course materials, type of assessment, course level and study design [15].

Although the flipped classroom strategy in medical programs has been evaluated extensively, few studies have focused on this approach in undergraduate medical laboratory sciences or during the internship year [16]. Thus, this study focused on this strategy in an undergraduate hematology course in a medical laboratory sciences program. The course integrated physiology, pathophysiology and clinical manifestations into the teaching of expected laboratory diagnoses. Learners were encouraged to focus on a topic with the required objectives and learning outcomes that included knowledge and reasoning in clinical laboratories. The flipped classroom was modified by the introduction of a case study discussion session after the educational contents and objectives were taught to students. The aim of this study was to compare the traditional teaching strategy with student-centred learning strategies implemented through the flipped classroom.

## Materials and methods

Two cohorts of third-year students of medical laboratory science participated in the study. The study was designed to suit the roles and regulations applied in Fakeeh College for Medical

Sciences (FCMS) and was approved by medical education departments at FCMS., Cohort 1 included 30 students (27 students were female and 3 students were males) enrolled in the hematology course during the academic year 2018–2019, which was taught according to the traditional teaching strategy, in which a 50-minute lecture was presented, and the students gave full attention to the instructor. Cohort 2 included 24 students (20 students were female and 4 students were males) enrolled in the course during the academic year 2019–2020, part of which was taught according to the flipped classroom strategy.

In the course for cohort 2, the flipped classroom was the teaching method for the theoretical part of the course, which is considered half of the course topics. In contrast, the other half, the practical sessions, were conducted in dedicated teaching, demonstration and performing experiments. The flipped classroom strategy was introduced to students in cooperation at the beginning of the semester. The sessions were conducted as published timetable and piloted in three phases: pre-class, in-class and after-class activities. The pre-class phase involved upload-ing the study materials, learning objectives and other supporting materials that helped the stu-dents prepare for the in-class sessions. The in-class part of the flipped classroom was divided into four parts: answering students' questions and clarifications, free discussion with col-leagues, post-examination discussion and a brief lecture focusing on the main objectives and giving feedback to students. For in-class activities, the course instructor, with other course facilitators, divided the students into five groups and presented a unified case study that cov-ered the intended learning outcomes. Each group was given 5 minutes to clarify the given case study and to ask the instructor questions. Thereafter, the students in each group discussed the case study to obtain information that corresponded to the intended learning outcomes. This discussion lasted for approximately 45 minutes. During the discussion time, the instructors supported the students in the discussion and guided the groups to follow the topic objectives and achieve intended learning outcomes.

In the post-class phase, students in cohort 2 were administered a formative assessment that was a mixture of multiple-choice (MCQs) and true-or-false questions. The formative was con-ducted on an electronic platform (Blackboard) and graded automatically after submitting the answer; thereafter, the instructor presented a review lecture on the corresponding topic with the feedback about points raised during the in-class discussion and formative questions.

At the end of the course, both cohorts took a final theoretical examination composed of 40 MCQs and matching questions based on the objectives presented at the beginning of the rela-tive academic year. The achievement in intended learning outcomes was analyzed by calculat-ing the difficulty index for each question by dividing the total scored mark of the correct answer in each question by the highest possible mark of the question multiplied by the total number of students. To assess the effectiveness of the flipped classroom on students' perfor-mance, the mean score on the final examinations for the flipped classroom in the course was compared with the mean score for cohort 1. In addition, the mean grade-point averages of the two cohorts were compared to evaluate the variation in academic performance. A descriptive statistical analysis, including overall grade, mean of knowledge achievement score and mean of cognitive achievement score, was used to assess statistical significance through unpaired T-test. A $p$ value of $<0.05$ was considered statistically significant using SAS version 9.3 (SAS Institute, Cary, NC, USA).

To assess the students' perceptions about the flipped classroom, a survey was distributed to the students at the end of academic year 2019–2020. The questions on the survey concerned the effectiveness of the flipped classroom in helping students gain knowledge and improve cognitive thinking [17]. The survey was distributed to students electronically, and their answers were on a 5-point Likert-type scale (5 = strongly agree, 4 = agree, 3 = neutral, 2 = dis-agree and 1 = strongly disagree). The ethical approval was obtained according to applied policy

and procedures. The participants were informed about the process and considered their contribution to the survey as informed consent. All the participants were ensured of the anonymity of their responses and given feedback. Moreover, the input and their perception about the flipped class sessions were conducted after the course completion to ensure their marks would not be affected by their decision

## Results

Women made up 90% of cohort 1 and 83% of cohort 2 (Table 1). The mean grade in cohort 1 was significantly lower than that in cohort 2 ($p$ = 0.0193) (Table 2). Of the students in cohort 1, 80% passed the course; of those in cohort 2, 100% passed (Fig 1A). Of the students in cohort 1, six received a grade of A, seven received a B, three received a C, eight received a D and six received an F. Of those in cohort 2, on the other hand, six students received an A, 10 received a B, seven received a C and one received a D, but none failed. Overall, however, the grade-point averages (GPA) for cohort 1 (3.92) and cohort 2 (3.94) did not differ significantly ($p$ = 0.459).

In Table 2, the course learning outcomes, which were divided into two domains—main knowledge and cognitive—cohort 2 showed a statistically significant improvement over cohort 1 in both domains ($p$ < 0.0001; Fig 1B and 1C).

With regard to the survey about perception of the flipped classroom as a teaching strategy in the hematology course, 100% of students in cohort 2 responded. For consistency and reliability of items in the survey, Cronbach's alpha was 0.991 (Table 3). Approximately 80% of students agreed that they had a better understanding of the material and better learning skills and that they achieved the required knowledge and skills in the field. However, 8% of students preferred the traditional approach over the flipped classroom approach, and almost 41% of students believed that the practical sessions should be conducted in flipped classroom sessions (Table 4). The main concern raised by students was the time necessary to prepare and study for this type of teaching. Nearly 83% of students agreed that such sessions required more time to prepare for the class and for the instructor to teach. Finally, most of the students agreed that the flipped classroom sessions improved their logical thinking and provided more information in the field that helped them improve their performance throughout the semester.

## Discussion

The National Centre for Academic Accreditation and Evaluation of the Education & Training Evaluation Commission in Saudi Arabia recommended using e-learning and blended teaching strategies. Studies of student-centred teaching methods in medical laboratory undergraduate courses are rare [18]. The main purpose of this study was to assess the difference between the flipped classroom—a student-centred learning method—and a didactic teaching strategy in an undergraduate hematology course in a medical laboratory sciences program by evaluating students' academic performance and their satisfaction with the flipped classroom method. The results suggested that the flipped classroom method had a positive influence on learning outcomes achievement and on learning experience in comparison with the traditional teaching method.

**Table 1. Distribution of Students based on gender.**

| Gender | Cohort 1 (n = 30) | Cohort 2 (n = 24) |
|---|---|---|
| Female | 90% (27) | 83.33%(20) |
| Male | 10% (3) | 16.67%(4) |
| Total | 100% (30) | 100%(24) |

**Table 2. Course grade and leaning outcomes achievements among two cohorts.**

| Group | Grade Achievement [mean (SD)] | Knowledge Achievement [mean (SD)] | Cognitive Achievement [mean (SD)] | Overall GPA in the Program |
|---|---|---|---|---|
| Cohort 1 | 72.93 (15.15) | 52.26 (11.7) | 50.00 (5.28) | 3.92 |
| Cohort 1 | 80.71 (7.93) | 82.55 (10.6) | 82.45 (6.47) | 3.94 |
| *p* value | < 0.0193 | < 0.0001 | < 0.0001 | 0.459 |

The examination scores for cohort 2 (the students who experienced the flipped classroom) were better than those for cohort 1. The grade-point averages for cohorts 1 and 2 were similar, which indicates that the academic performances of the two cohorts should not have affected the reliability of the results of this study. In addition, both cohorts were presented with the same course materials, test blueprints and assessment methods in the hematology course.

Although the flipped classroom strategy positively affected student achievement in most studies that focused on subjects in the medical field, few studies have focused on medical laboratory courses [16,18–20]. Huang et al. found that the achievement of research objectives was significantly better among interns who attended flipped classroom sessions than those who attended traditional classes [16]. Regarding hematology courses, however, two other studies were conducted in medical programs and showed mixed results [14,21]. Porcaro et al. showed that grades on the final examination improved in the flipped classroom and that the students recommended continually using this teaching strategy in the course [21], whereas Sajid et al. revealed no difference in the students' grades but did find that students were strongly satisfied with the teaching strategy [14]. In an undergraduate speciality in medical laboratory sciences, this study showed that the flipped classroom and the use of student-centred methods resulted

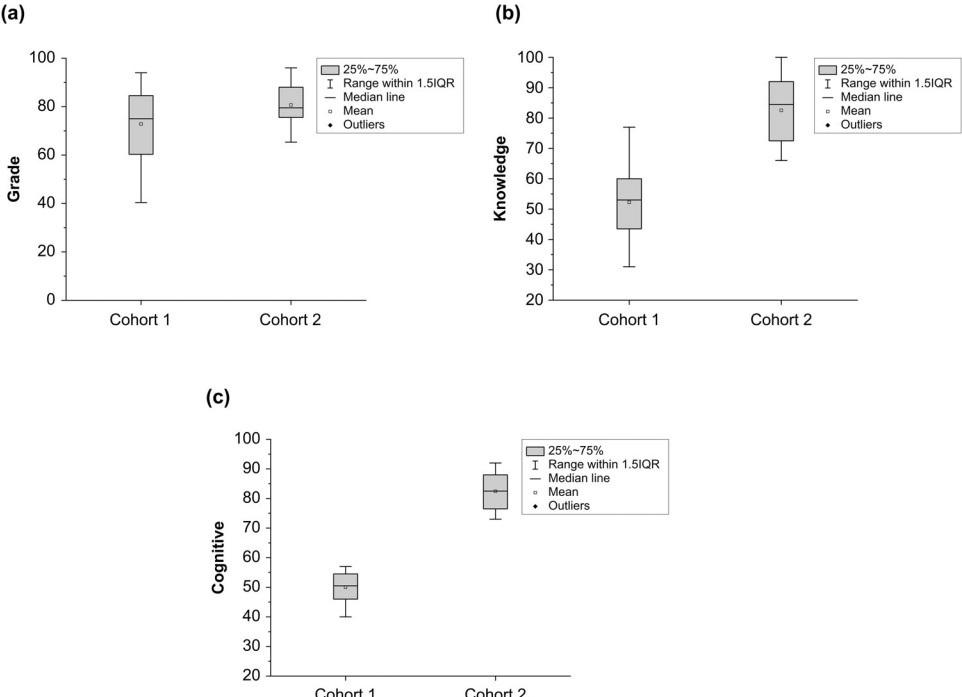

**Fig 1. The analysis of students' achievement in the hematology course.** (a) The range of final grades in the course in cohorts 1 and 2. (b) The performance of students during the final examination in comparison with their achievement in the knowledge domain. (c) The performance of students during the final examination in comparison with their achievement in the cognitive domain. IQR, interquartile range.

**Table 3.  Cronbach's alpha analysis of survey items concerning students' perceptions about the flipped classroom teaching strategy.**

| Cronbach's alpha | Cronbach's alpha based on standardized items | Number of items |
|---|---|---|
| 0.989 | 0.991 | 15 |

in significantly improved academic performance and a better learning experience. This finding is consistent with that of a study conducted in a pathophysiology course for medical students [22] in which some students attended flipped classroom sessions and others attended sessions in which the traditional teaching approach was used by the same course instructor [22]. The examination scores in flipped classroom sessions were significantly better than those in traditional sessions [22].

The students in this study who attended flipped classroom sessions participated with enthusiasm and energy in the in-class discussions. The low-performing students participated more actively than did low-performing students who attended traditional sessions. This finding indicates that active-learning methods are probably appropriate for all students, regardless of their academic performance. In addition, students were able to read and prepare for course topics at the appropriate times and to overcome difficulties with English, which was used in class and was a second language for most of the students. Herrero and Quiroga demonstrated an improvement in the grades of poor-performing students in the flipped classroom sessions, which suggested that this strategy helped those students both overcome obstacles in understanding the main objectives and prepare to discuss with a course instructor the areas that need to be clarified [22].

The students in this study responded positively to the implementation of the flipped classroom and agreed that it improved their performance. However, students stated that flipped classroom sessions required more time to search for and read about assigned topics and that this could affect their performance in other courses. In addition, half the students agreed that flipped classroom sessions should be continued; whereas 41.67% were neutral and 8.33% disagreed that it should be used as teaching strategy instead of the traditional teaching method (Table 2). In a similar study, students in medical school indicated that the flipped classroom sessions improved their understanding of course material and helped them accomplish the required objectives for the topics, in comparison with traditional teaching [14,16,23].

However, this study revealed some disadvantages of the flipped classroom. First, it was difficult to conduct each session in 50 minutes, especially in as much as the flipped classroom was divided into four parts. Moreover, directing and guiding student groups and assessing the performance of each student during the session were challenging for one instructor.

## Limitations and future directions

In this study, the flipped classroom method did not cover all the course modules and activities. The study covered part of the theoretical material; the other theoretical and practical materials were taught through demonstration of the required skills and procedures, and at the end of the semester, the students were assessed through practical activities and written examinations. Thus the effectiveness of the flipped classroom method, integrated with discussion sessions and other student-centred learning methods, may be influenced by the practical material, but that was not studied. Furthermore, the number of students participating in the study was small; a larger number of participants is necessary to validate the results. Finally, MCQs were used on final examinations to assess students' performance, but questions that require longer answers, such as essays, may better reflect cognitive improvement.

**Table 4. Survey about students' perceptions of the flipped classroom as a teaching strategy (n = 24).**

| Statement | Strongly agree | Agree | Neutral | Disagree |
|---|---|---|---|---|
| Flipped classroom session provides better understanding of subject and learning skills | 29.17% (7) | 54.17%(13) | 16.67% (4) | 0.00% (0) |
| Flipped classroom session enhance students intellectual curiosity | 20.83% (5) | 45.83%(11) | 33.33% (8) | 0.00% (0) |
| Flipped classroom session give knowledge and skills that are helpful in field practice | 37.50% (9) | 45.83%(11) | 16.67% (4) | 0.00% (0) |
| Flipped classroom session help in better retaining of the subject | 33.33% (8) | 45.83%(11) | 20.83%(5) | 0.00%(0) |
| Flipped classroom session is preferred over traditional teaching | 20.83% (5) | 29.17%(7) | 41.67%(10) | 8.33%(2) |
| Flipped classroom session should include laboratory exercises | 41.67% (10) | 29.17%(7) | 25.00%(6) | 4.17%(1) |
| Flipped classroom session should have allotted more time for each topic | 41.67% (10) | 41.67%(10) | 16.67%(4) | 0.0%(0) |
| Flipped classroom session topics related to same semester is preferred | 45.83% (11) | 12.50%(3) | 37.50%(9) | 4.17%(1) |
| Flipped classroom session should be in the form of case discussions | 45.83% (11) | 33.33%(8) | 20.83%(5) | 0.00%(0) |
| Flipped classroom session reduces the amount of time needed for study when compared to lectures | 33.33% (8) | 25.00%(6) | 33.33%(8) | 4.17%(1) |
| Flipped classroom session improves the application of knowledge | 45.83% (11) | 25.00%(6) | 29.17%(7) | 0.00%(0) |
| Flipped classroom session develops logical thinking | 45.83% (11) | 20.83%(5) | 20.83%(5) | 8.33%(2) |
| Flipped classroom session provides extra information | 41.67% (10) | 29.17%(7) | 16.67%(4) | 12.50%(3) |
| Flipped classroom session requires a long time for preparation and conduction | 45.83% (11) | 20.83%(5) | 33.33%(8) | 0.00%(0) |

## Conclusion

Implementing student-centred learning systems has become necessary in medical education to improve the quality of education. The flipped classroom is a beneficial and powerful teaching strategy in that it helps students gain desirable knowledge and improves life-long learning skills. Student-centred active learning can also improve retention of material, as demonstrated on standard examinations.

## Acknowledgments

The author thanks Dr Hani Elsayed, associate professor at Fakeeh College of Medical Sciences, Jeddah, Saudi Arabia, for supporting in analysing the data, and Dr Amira Ismael, assistant professor in the Medical Education Department, Fakeeh College of Medical Sciences, for reviewing this article and providing valuable feedback.

## Author Contributions

**Investigation:** Haitham Qutob.

**Writing – original draft:** Haitham Qutob.

**Writing – review & editing:** Haitham Qutob.

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
