## [Decision Letter · Decision Letter 0]

24 Feb 2022

PONE-D-21-37375Effect of flipped classroom approach in the teaching of a haematology coursePLOS ONE

Dear Dr. Qutob,

Thank you for submitting your manuscript to PLOS ONE. After careful consideration, we feel that it has merit but does not fully meet PLOS ONE’s publication criteria as it currently stands. Therefore, we invite you to submit a revised version of the manuscript that addresses the points raised during the review process. Based on the review results, a minor revision has been suggested before making final decision. Please respond to the reviewers' comments and revise the manuscript accordingly. 

We look forward to receiving your revised manuscript.

Kind regards,

Di Zou

Academic Editor

PLOS ONE

Journal Requirements:

a) Did participants provide their written or verbal informed consent to participate in this study?

Reviewers' comments:

Reviewer's Responses to Questions

**Comments to the Author**

1. Is the manuscript technically sound, and do the data support the conclusions?

Reviewer #1: Yes

Reviewer #2: No

Reviewer #3: Yes

2. Has the statistical analysis been performed appropriately and rigorously? 

Reviewer #1: I Don't Know

Reviewer #2: Yes

Reviewer #3: Yes

3. Have the authors made all data underlying the findings in their manuscript fully available?

Reviewer #1: Yes

Reviewer #2: Yes

Reviewer #3: Yes

4. Is the manuscript presented in an intelligible fashion and written in standard English?

Reviewer #1: Yes

Reviewer #2: Yes

Reviewer #3: Yes

5. Review Comments to the Author

Reviewer #1: Description of study demographics and statistical methods should be described, in detail, in the Methods section. Saying "A p value of <0.05 was considered statistically significant according to calculations using SAS version 9.3 (SAS Institute, Cary, NC, USA)" is not sufficient. What method was used? What calculations> How can the reader understand what was done? Shouldn't all the p values be in the Table? Authors says number of students is a limitation (certainly it was, the n value is extremely small), but that is also complicated by the extreme female sex-bias in the sample. At the very least, another limitation is that it is unknown if a more sex-balanced, or male sex-biased, sample would yield similar results. Please clarify these issues.

Reviewer #2: Effect of flipped classroom approach in the teaching of a haematology course

Article # PONE-D-21-37375

Methodology

Cohort 1 included 30 students enrolled in the hematology course during the 129 academic year 2018–2019, which was taught according to the traditional teaching strategy, in which a 50-minute lecture was presented, and the students gave full attention to the instructor. Cohort 2 included 24 students enrolled in the course during the academic year 2019–2020, part of which was taught according to the flipped classroom strategy.

In the course for cohort 2, the flipped classroom was the teaching strategy for half of the course topics; for the other half, the course was conducted according to traditional didactic

This is not clear how students were recruited for participation in lecture session and Flipped session 2nd, it is also not clear how many topics of hematology course were flipped , 1, 2 or 3 etc.

Line 149 In the post-class phase, students were administered a quiz that was a mixture of multiple choice (MCQs) and true-or-false questions.

It means, formative assessment was done immediately after the Flipped session. Was it done for lecture session, it is not clear

Acknowledgement

Line 296-9

The author thanks Dr Hani Elsayed, associate professor at Fakeeh College of Medical Sciences, 297 Jeddah, Saudi Arabia, for supporting and analysing the data, and Dr Amira Ismael, assistant professor in the Medical Education Department, Fakeeh College of Medical Sciences, for reviewing this article and providing valuable feedback.

Dr Hani Elsayed and Dr Amira Ismael should be enlisted as co- authors ( according to guidelines of authorship)

Reviewer #3: This study presents findings from a comparison of flipped classroom versus lecture based teaching in an undergraduate hematology course. In one year a class session was delivered by lecture while for the cohort the following year it was delivered by flipped class with an active learning session. An improvement in performance in final exam was found in with the flipped class and students perceptions of the experience were generally positive.

Specific comments:

- This study is limited as it parallels what has already been presented in multiple existing studies. Additional detail would have to be provided to demonstrate why the subject material here differentiates this from similar studies and what novelty is added to the literature.

- A very small number of students is studied – 30 compared with 24. There is a dramatic difference with 6 students receiving an F in the lecture group and no students receiving an F in the flipped class group. It is difficult to determine from the description provided how much of the course was converted to flipped class. It looks like it was only a single session, therefore it is hard to comprehend that this was responsible for the dramatic change. Repeating the study and increasing the number of participants would solidify this data.

- In relation to the comment above - the author needs to provide more detail on the study design – both a clear step by step comparison for the lecture vs flipped session plus assessments as well as how these sessions were incorporated into the whole course and how much of the final exam related to this session.

6. PLOS authors have the option to publish the peer review history of their article (what does this mean?). If published, this will include your full peer review and any attached files.

Reviewer #1: No

Reviewer #2: No

Reviewer #3: No

---

## [Author Response · Author response to Decision Letter 0]

9 Mar 2022

I. Editor's Comments:

I.1. Please amend your current ethics statement to address the following concerns:

a) Did participants provide their written or verbal informed consent to participate in this study?

It was verbal informed consent as it was apart of the orientation given to the students about the delivery of flipped classrooms, In addition to that the confidentiality of information was kept as the questionnaires distrusted to students were anonymous.

The ethical approval was obtained according to applied policy and procedures. The participants were informed about the process and considered their contribution to the survey as informed consent. All the participants were ensured of the anonymity of their responses and given feedback. Moreover, the input and their perception about the flipped class sessions were conducted after the course completion to ensure their marks would not be affected by their decision

I.2. Please review your reference list to ensure that it is complete and correct. If you have cited papers that have been retracted, please include the rationale for doing so in the manuscript text, or remove these references and replace them with relevant current references. Any changes to the reference list should be mentioned in the rebuttal letter that accompanies your revised manuscript. If you need to cite a retracted article, indicate the article's retracted status in the References list and also include a citation and full reference for the retraction notice.

All were checked, and Ref number 4 was corrected "Simon FA, Aschenbrener CA. Undergraduate medical education accreditation as a driver of lifelong learning. J Contin Educ Health Prof. 2005 Summer;25(3):157-61. doi: 10.1002/chp.23. PMID: 16173065."

I.3. Can you please upload an additional copy of your revised manuscript that does not contain any tracked changes or highlighting as your main article file. This will be used in the production process if your manuscript is accepted. Please amend the file type for the file showing your changes to Revised Manuscript w/tracked changes. Please follow this link for more information

It is attached and named as “revised manuscript without track Changes”

The tracked file is named as “Revised Manuscript with Track Changes”

I.4. Please ensure that you refer to Table 4 in your text as, if accepted, production will need this reference to link the reader to the Table.

Done

I.5. Thank you for including your ethics statement on the online submission form: 

"The study has been approved by Medical Education board at Fakeeh College for Medical Sciences". 

To help ensure that the wording of your manuscript is suitable for publication, would you please also add this statement at the beginning of the Methods section of your manuscript file.

It is added in the method in the second line of the fist paragraph.

II. Reviewer #1: 

II.1. Description of study demographics and statistical methods should be described, in detail, 

Cohort 1 included 30 students (27 students were female and 3 students were males) enrolled in the haematology course during the academic year 2018–2019, which was taught according to the traditional teaching strategy, in which a 50-minute lecture was presented, and the students gave full attention to the instructor. Cohort 2 included 24 students (20 students were female and 4 students were males) enrolled in the course during the academic year 2019–2020, part of which was taught according to the flipped classroom strategy.

Table 1 is added

II.2. in the Methods section. Saying "A p value of <0.05 was considered statistically significant according to calculations using SAS version 9.3 (SAS Institute, Cary, NC, USA)" is not sufficient. 

A descriptive statistical analysis, including overall grade, mean of knowledge achievement score and mean of cognitive achievement score, was used to assess statistical significance through unpaired T-test.

One table is inserted to present the difference between the two cohort in students' mean score of knowledge and cognitive domain.

II.3. What method was used?

Done, as there were two different cohort, Independent sample T-test was utilized to show the significance of the study

II.4. What calculations> How can the reader understand what was done?

It is explained in lines 176 to 180. The statistical method was added

II.5. Shouldn't all the p values be in the Table?

One table is presented to show the significant if t-test

II.6. Authors says number of students is a limitation (certainly it was, the n value is extremely small), but that is also complicated by the extreme female sex-bias in the sample. 

At the very least, another limitation is that it is unknown if a more sex-balanced, or male sex-biased, sample would yield similar results. Please clarify these issues.

From author point of view , sex- biased here is already comparable as there is small percentage of male students in both cohort (4 students in cohort 1 and 3 in another cohort and this make the two cohort in more homogenous and actually this bias is non controllable as it is related to the cohort and the number of male students in the batch.

III. Reviewer #2: 

Methodology

III.1. Cohort 1 included 30 students enrolled in the hematology course during the 129 academic year 2018–2019, which was taught according to the traditional teaching strategy, in which a 50-minute lecture was presented, and the students gave full attention to the instructor. Cohort 2 included 24 students enrolled in the course during the academic year 2019–2020, part of which was taught according to the flipped classroom strategy.

In the course for cohort 2, the flipped classroom was the teaching strategy for half of the course topics; for the other half, the course was conducted according to traditional didactic

This is not clear how students were recruited for participation in lecture session and Flipped session 2nd, it is also not clear how many topics of hematology course were flipped , 1, 2 or 3 etc.

In the course for cohort 2, the flipped classroom was the teaching method for the theoretical part of the course, which is considered half of the course topics. In contrast, the other half, the practical sessions, were conducted in dedicated teaching, demonstration and performing experiments. The flipped classroom sessions were conducted as published timetable and piloted in three phases: pre-class, in-class and after-class activities.

III.2. Line 149 In the post-class phase, students were administered a quiz that was a mixture of multiple choice (MCQs) and true-or-false questions. It means, formative assessment was done immediately after the Flipped session. Was it done for lecture session, it is not clear

In the post-class phase, students in cohort 2 were administered a formative assessment that was a mixture of multiple-choice (MCQs) and true-or-false questions. The formative was conducted on an electronic platform (Blackboard) and graded automatically after submitting the answer; thereafter, the instructor presented a review lecture on the corresponding topic with the feedback about points raised during the in-class discussion and formative questions. 

Acknowledgement

III.3. Line 296-9

The author thanks Dr Hani Elsayed, associate professor at Fakeeh College of Medical Sciences, 297 Jeddah, Saudi Arabia, for supporting and analyzing the data, and Dr Amira Ismael, assistant professor in the Medical Education Department, Fakeeh College of Medical Sciences, for reviewing this article and providing valuable feedback.

Dr Hani Elsayed gave me the SAS software and showed me how I could use it to conduct the analysis. Dr Amira Ismael reviewed the article from the point of medical education view and expert in reviewing reports before sending them to a journal for submission to minimize the comments. The College requests this process. 

IV. Reviewer #3:

This study presents findings from a comparison of flipped classroom versus lecture based teaching in an undergraduate hematology course. In one year a class session was delivered by lecture while for the cohort the following year it was delivered by flipped class with an active learning session. An improvement in performance in final exam was found in with the flipped class and students perceptions of the experience were generally positive.

Specific comments:

IV.1. This study is limited as it parallels what has already been presented in multiple existing studies. Additional detail would have to be provided to demonstrate why the subject material here differentiates this from similar studies and what novelty is added to the literature.

I agree that many studies are conducted to assess the effectiveness of flipped classrooms. However, few studies were focused on the medical laboratory field and hematology course as one of the main subjects in the area. In Saudi Arabia, these new strategies are introduced to MBBS programs, and few institutions started to introduce them to undergraduate nursing programs. Such studies in specific medical professions could encourage other colleagues and coordinators to integrate student-centered learning in their teaching strategy. So, the novelty in this study is related to the context and the course introduced, as to our knowledge this is the first study to apply flipped classes as method of teaching in MLS context

IV.2. A very small number of students is studied – 30 compared with 24. There is a dramatic difference with 6 students receiving an F in the lecture group and no students receiving an F in the flipped class group. 

It is difficult to determine from the description provided how much of the course was converted to flipped class. It looks like it was only a single session, therefore it is hard to comprehend that this was responsible for the dramatic change. Repeating the study and increasing the number of participants would solidify this data.

I agree, and it is mentioned and suggested in the study limitation. In addition, the number of enrolled students to the program is limited to around 25.

IV.3. In relation to the comment above - the author needs to provide more detail on the study design – both a clear step by step comparison for the lecture vs flipped session plus assessments as well as how these sessions were incorporated into the whole course and how much of the final exam related to this session.

It is mentioned in the methods:

In the course for cohort 2, the flipped classroom was the teaching method for the theoretical part of the course, which is considered half of the course topics. In contrast, the other half, the practical sessions, were conducted in dedicated teaching, demonstration and performing experiments. The flipped classroom strategy was introduced to students in cooperation at the beginning of the semester. The sessions were conducted as published timetable and piloted in three phases: pre-class, in-class and after-class activities.

At the end of the course, both cohorts took a final theoretical examination composed of 40 MCQs and matching questions based on the objectives presented at the beginning of the relative academic year. The achievement in intended learning outcomes was analyzed by calculating the difficulty index for each question by dividing the total scored mark of the correct answer in each question by the highest possible mark of the question multiplied by the total number of students.

---

## [Decision Letter · Decision Letter 1]

4 Apr 2022

Effect of flipped classroom approach in the teaching of a haematology course

PONE-D-21-37375R1

Dear Dr. Qutob,

We’re pleased to inform you that your manuscript has been judged scientifically suitable for publication and will be formally accepted for publication once it meets all outstanding technical requirements.

Kind regards,

Di Zou

Academic Editor

PLOS ONE

Additional Editor Comments (optional):

Reviewers' comments:

Reviewer's Responses to Questions

**Comments to the Author**

1. If the authors have adequately addressed your comments raised in a previous round of review and you feel that this manuscript is now acceptable for publication, you may indicate that here to bypass the “Comments to the Author” section, enter your conflict of interest statement in the “Confidential to Editor” section, and submit your "Accept" recommendation.

Reviewer #1: All comments have been addressed

Reviewer #2: All comments have been addressed

Reviewer #3: All comments have been addressed

2. Is the manuscript technically sound, and do the data support the conclusions?

Reviewer #1: Yes

Reviewer #2: Yes

Reviewer #3: Yes

3. Has the statistical analysis been performed appropriately and rigorously? 

Reviewer #1: Yes

Reviewer #2: Yes

Reviewer #3: Yes

4. Have the authors made all data underlying the findings in their manuscript fully available?

Reviewer #1: Yes

Reviewer #2: Yes

Reviewer #3: Yes

5. Is the manuscript presented in an intelligible fashion and written in standard English?

Reviewer #1: Yes

Reviewer #2: Yes

Reviewer #3: Yes

6. Review Comments to the Author

Reviewer #1: It appears that my comments have been addressed; therefore, the paper can be accepted. I have no other comments.

Reviewer #2: authors have revised the manuscript and previously all the points / comments he/she has fulfilled

Reviewer #3: Comments were addressed.

There is a typo in Table 2 - both rows are labeled as Cohort 1 - the second should be Cohort 2?

7. PLOS authors have the option to publish the peer review history of their article (what does this mean?). If published, this will include your full peer review and any attached files.

Reviewer #1: No

Reviewer #2: No

Reviewer #3: No

---

## [Editor Report · Acceptance letter]

13 Apr 2022

PONE-D-21-37375R1 

Effect of Flipped Classroom Approach in The Teaching of A Hematology Course 

Dear Dr. Qutob:

I'm pleased to inform you that your manuscript has been deemed suitable for publication in PLOS ONE. Congratulations! Your manuscript is now with our production department. 

Kind regards, 

on behalf of

Dr. Di Zou 

Academic Editor

PLOS ONE